# Effects of MAPK Homologous Genes on Chemotaxis and Egg Hatching in *Meloidogyne incognita*

**DOI:** 10.3390/pathogens14121290

**Published:** 2025-12-16

**Authors:** Youjing Wang, Mingxin Liu, Jiefang Li, Caiwei Hu, Yajun Liu

**Affiliations:** State Key Laboratory for Conservation and Utilization of Biological Resources in Yunnan Province, Kunming 650032, China

**Keywords:** *Meloidogyne incognita*, MAPK pathway, chemotaxis, eggs hatching, signal-transduction

## Abstract

Root-knot nematodes, known for their broad host range and the challenges associated with their control, cause significant losses in global agricultural productivity. The mitogen-activated protein kinase (MAPK) pathway amplifies signals through a phosphorylation cascade to regulate downstream transcription factors or effector proteins, which is essential for maintaining normal physiological functions in nematodes. This study presents a comprehensive characterization of the MAPK signaling cascade regulatory network in *Meloidogyne incognita*, elucidating the regulatory roles of *Mi-let-60*, *Mi-lin-45*, *Mi-mek-2*, and *Mi-mpk-1* in host chemotaxis and egg hatching behavior. Importantly, RNAi-mediated silencing of *Mi-mpk-1* resulted in a 17-fold increase in egg hatching rate and significantly impaired chemotactic responses while silencing of *Mi-let-60* led to a marked reduction in root gall formation and egg mass production. These results not only confirmed the evolutionary conservation of this pathway but also identified a feedback regulation mechanism unique to parasitic nematodes. These findings provide valuable insights for future research on signal transduction pathways and the sustainable management of root-knot nematodes.

## 1. Introduction

Root-knot nematodes (*Meloidogyne* spp.) are pathogenic organisms responsible for root-knot disease in crops. They possess an exceptionally broad host range, are widely distributed in tropical and subtropical regions, and are capable of infecting over 3000 plant species, resulting in substantial annual economic losses to global crop production [1]. Current management strategies for *Meloidogyne* spp. primarily encompass chemical control, physical control, biological control, agricultural practices, and improved field management practices. However, each of these approaches has inherent limitations, necessitating the development of novel and more effective control methods.

The MAPK pathway has been well studied in yeast, mammals and plants. Mitogen-activated protein kinases (MAPKs) are serine/threonine-specific protein kinases. MAPK is one of the most evolutionarily conserved signal transduction pathways and regulates a wide range of physiological processes, including cell proliferation, differentiation, movement and survival [2]. In *Caenorhabditis elegans*, the RTK/Ras/MAPK signaling cascade is the most thoroughly studied pathway. MAPK signaling is a major regulator of key meiotic events in *C*. *elegans* [3,4]. This pathway includes the signaling molecules LET-60 (Ras), LIN-45 (Raf), and MEK-2 (Mek), which transmit extracellular signals, leading to the phosphorylation and activation of its terminal member MPK-1 (Erk) [5,6].

The role of the RTK/Ras/MAPK signaling pathway in regulating chemotactic behavior in *C. elegans* and *Meloidogyne* spp. remains poorly documented. The MAPK pathway modulates olfactory perception in *C. elegans*. Mutations in genes *let-60*, *lin-45*, and *mek-2* resulted in varying degrees of sensory deficits in response to isoamyl alcohol, diacetyl, and trimethylthiazole. Furthermore, immunohistochemical experiments indicated that isoamyl alcohol activates the Ras-MAPK pathway and regulates sensory signal transduction within olfactory neurons [7]. Additionally, the G protein-coupled receptor SRA-13 has been shown to inhibit the perception of isovaleric acid and acetoin via GPA-5-mediated suppression of the Ras-MAPK pathway, suggesting functional crosstalk between G protein-coupled receptor signaling and the Ras-MAPK pathway in olfactory processing [8]. Mutants of the Ras-MAPK pathway exhibit defects in early-stage adaptation, revealing a dual role for this pathway in sensory processing within the *C. elegans* nervous system. Specifically, Ras-MAPK activity in olfactory neurons is crucial for odor perception, whereas its activity in the AIY interneurons—which receive synaptic input from olfactory neurons—is essential for odor adaptation [9]. Another study indicated that RGEF-1b activates the LET-60 (Ras)–MAPK-1 (ERK) signaling cascade in AWC neurons to mediate chemotaxis toward volatile odorants [10]. Genetic interaction analyses further demonstrated that the JNK-family MAPK KGB-1 functions downstream of two signaling pathways to regulate salt chemotaxis learning. Moreover, the NSY-1/SEK-1 pathway operates in sensory neurons ASH, ADF, and ASER to modulate learned salt chemotaxis. The neuropeptide NLP-3, expressed in ASH, ADF, and ASER neurons, and its receptor NPR-15, expressed in AIA interneurons that receive synaptic input from these sensory neurons, function in a manner consistent with NSY-1/SEK-1 signaling. These findings suggest that this MAPK pathway may influence neuropeptide signaling between sensory neurons and interneurons, thereby facilitating high-salt chemotaxis following conditioning [11].

MPK-1 signaling regulates germ cell apoptosis by restricting CEP-1 protein expression to late pachytene cells. The frequency of alternative transcript selection (ATS) classes shifts in a stepwise manner as stem cell daughters initiate differentiation, suggesting that ATS regulation may serve as a key mechanism of developmental gene control [12]. In mutants with elevated levels of activated MPK-1, oocyte quality declines more rapidly compared to wild-type worms, whereas reduced MPK-1 activity is associated with improved oocyte quality. These findings demonstrate that MAPK signaling governs germline aging and may represent a potential target for mitigating the age-related decline in oogenesis [13]. A recent study has shown that the evolutionarily conserved c-Jun N-terminal kinase-like MAPK pathway is essential for dauer formation in *C. elegans* under environmental stress. Loss-of-function mutations in the MLK-1–MEK-1–KGB-1 MAPK cascade suppress entry into the dauer developmental stage, highlighting its pivotal role in stress-induced developmental plasticity [14].

Research on RTK/Ras/MAPK-related genes in *Meloidogyne* spp. remains limited. The mitogen-activated protein kinase gene *Me-mapk1* was cloned from *M. enterolobii*, and RNA interference-mediated silencing of this gene significantly impaired nematode infectivity [15]. Similarly, Wang [16] cloned the mapk gene from the pine wood nematode *Bursaphelenchus xylophilus*; in vitro RNAi experiments revealed an approximately 50% reduction in reproductive capacity, suggesting a potential role of MAPK in nematode reproduction. The *let-60* gene expressed across multiple developmental stages of *M. incognita* (eggs, second-stage juveniles, and parasitic stages), indicating its possible regulatory function in hatching and parasitism. Following in vitro RNAi-mediated silencing of *let-60*, a marked decrease was observed in both host plant infection ability and reproductive capacity. Significant reductions were recorded in root gall formation, egg mass production, total egg count, and hatching rate compared to controls, confirming the essential role of *let-60* in parasitic success [17]. In vitro soaking RNAi targeting *Mi-mpk-1* in eggs and second-stage juveniles of *M. incognita* resulted in a 17-fold increase in egg hatching rate and elevated juvenile mortality relative to the control. When treated juveniles were inoculated onto tomato roots, a significant reduction in root galling and egg mass formation was observed, demonstrating that *Mi-mpk-1* is crucial for nematode development and parasitism [18]. Dong [19] applied RNA interference to reduce mpk-1 expression in *M. incognita* by approximately 33%. At 35 days post-inoculation, root gall numbers decreased by 32%, while egg masses and total eggs were reduced by 42% and 22%, respectively.

Chemotaxis constitutes the initial phase of host recognition and localization by J2s of root knot nematodes, facilitating subsequent root invasion. Chemical signals perceived by plant-parasitic nematodes are often derived from the microenvironments established by host plants to support their own growth. For example, carbon dioxide (CO_2_) released by root systems during respiration dissolves in the soil, generating a weakly acidic rhizosphere that exhibits chemotactic attraction toward nematodes [20]. In addition to biotic factors, abiotic factors such as specific ions, salt concentrations, and temperature gradients have also been shown to influence the chemotactic behavior of these organisms [21,22]. Notably, water-soluble compounds and volatile organic compounds (VOCs) present in root exudates are considered primary mediators of nematode chemotaxis [23]. Among these chemical cues, volatile compounds are generally understood to facilitate long-distance navigation of nematodes toward host roots, whereas water-soluble compounds are thought to guide precise positioning on the root surface and support the initiation of infection processes over shorter distances [24]. Concurrently, egg hatching is essential for the generation of infective J2s. This study focuses on MAPK genes exhibiting high sequence homology between *M. incognita* and *C. elegans* to investigate the functional role of the RTK/Ras/MAPK signaling pathway in regulating both egg hatching and chemotactic behavior in *M. incognita.* The results provide insight into the molecular mechanisms underlying chemotaxis and hatching signal transduction, thereby laying the groundwork for developing novel strategies to control *Meloidogyne* spp. infestations on critical early-life processes.

## 2. Materials and Methods

### 2.1. Culture of Tomato and of Meloidogyne incognita

Tomato seeds (*Lycopersicon esculentum*, purchased from Beijing Lvdongfang Agricultural Technology Institute) were surface-sterilized by sunlight exposure for 2 h and germinated on moist filter paper in 9 cm Petri dishes at 28 °C in darkness. Germinated seeds were transferred to seedling trays and grown to the four-leaf stage under controlled conditions. Plants were then transplanted into plastic pots (19 cm height × 9 cm diameter) filled with sterilized substrate (humus:vermiculite:fine sand = 7:2:1, *v*/*v*/*v*) and maintained in a greenhouse at 25 °C with 50% relative humidity under a 16/8 h light/dark cycle. Plants were watered every two days to maintain soil moisture at field capacity. The *Meloidogyne incognita* race 1 population was kindly provided by Dr. Chaojun Lu from the State Key Laboratory of Biological Resource Conservation and Utilization at Yunnan University. Species identification was confirmed by sequencing the ITS1-5.8S-ITS2 ribosomal DNA region. Second-stage juveniles (J2s) were collected from the maintained population using a Baermann funnel technique. After 11 days of acclimatization, each plant was inoculated with 2000 J2s.

### 2.2. RNA Interference of Genes Associated with Chemotaxis and Egg Hatching in Meloidogyne incognita

The cDNA sequences of *Meloidogyne incognita* genes *Mi-mpk-1* (GenBank: DQ923592.1), *Mi-mek-2* (GenBank: GU949544.1), *Mi-lin-45* (GenBank: GU949543.1), and *Mi-let-60* (GenBank: FJ176739.1) were obtained through homology alignment. Gene-specific primers for dsRNA synthesis and RT-qPCR were designed using Primer Premier 5.0, with T7 promoter sequences added to the 5′ end of dsRNA primers. Total RNA was extracted using the RaPure Total RNA Micro Kit (Magen , Guangzhou, China). Reverse transcription was performed with the PrimeScript RT reagent Kit with gDNA Eraser (Takara, Kusatsu, Shiga, Japan). Quantitative RT-qPCR was carried out using ChamQ Universal SYBR qPCR Master Mix (Vazyme, Nanjing, China), and relative gene expression levels were calculated via the 2^ΔΔCt^ method. For dsRNA production, target gene fragments were amplified, gel-purified with the E.Z.N.A. Gel Extraction Kit (Omega, Norcross, GA, USA), and cloned into plasmids using the E.Z.N.A. Plasmid DNA Mini Kit II (Omega). Plasmid sequencing was conducted by Tsingke Biotechnology (Beijing , China). dsRNA was synthesized in vitro using the MEGAscript RNAi Kit (Thermo Fisher Scientific, Waltham, MA, USA) and adjusted to a final concentration of 1 mg/mL. For primer information of RNAi for *Meloidogyne incognita*, see Appendix A. For RT-qPCR primers for *Meloidogyne incognita*, see Appendix A.

The collected J2s were centrifuged at 6000 rpm for 2 min and resuspended to 10,000 nematodes per 100 μL. For each replicate, 10,000–20,000 J2s were transferred to RNase-free 1.5 mL microcentrifuge tube, followed by the addition of 100 μL of 0.1% m-phenol, 50 μL of spermidine (30 mM), 10 μL of 5% gelatin, 100 μL of target dsRNA, and M9 buffer (comprising 5.8 g Na_2_HPO_4_·7H_2_O, 3.0 g KH_2_PO_4_, 5.0 g NaCl per liter, supplemented with 0.25 g MgSO_4_·7H_2_O in RNase-free water) to a final volume of 500 μL. The tubes were incubated on a rotary mixer (Kunming, Yunnan, China) for 24 h at room temperature. Three experimental groups were established: a Buffer control (without dsRNA), a Negative control (NC) with 500 bp non-target fragment from the MEGAscript™ RNAi Kit, and RNAi groups with target MAPK gene dsRNAs. Following RNAi treatment, the nematodes were divided for subsequent experiments: one portion for chemotaxis assays, another for inoculation tests, and the remainder for RT-qPCR analysis. The soaking RNAi protocol was adapted from published methods [18].

Egg masses were immersed in a 500 μL soaking solution containing 5% resorcinol (10 μL), 5% gelatin (5 μL), 30 mM spermidine (50 μL), dsRNA (1 mg/mL), and M9 buffer adjusted to a final volume of 500 μL, with approximately 50 egg masses per treatment. Control groups included sterile water and a dsRNA-free soaking solution. Hatching rates were recorded at 24 h, 48 h, and 72 h post-treatment. Following RNAi exposure, egg masses were transferred to 1.5 mL microcentrifuge tubes, washed three times with DEPC-treated water, rapidly frozen in liquid nitrogen, and stored at −80 °C for subsequent analysis.

### 2.3. Chemotaxis Assay of Meloidogyne incognita

Pluronic F-127 powder (23 g) was added to 80 mL of sterile water and dissolved at 4 °C for 2 h. Subsequently, 10 mM Tris and 10 mM morpholineethanesulfonic acid (MES) were added, and the volume was adjusted to 100 mL with sterile water. The solution was incubated at 4 °C for an additional 22 h to ensure complete dissolution of the gel, followed by pH adjustment to 7.0. The fully dissolved Pluronic F-127 gel was used to prepare citric acid solutions at a concentration of 0.1 M [25] which were stored at 4 °C for subsequent use. A 200 μL sterile pipette tip was cut to remove the distal 0.5 cm segment and filled with 30 μL of the prepared gel solution to serve as a dispensing unit. The gel was allowed to solidify at 28 °C. Dispensers without citric acid were prepared as blank controls. A 3-cm-diameter Petri dish was filled with 2–3 mL of the pre-prepared Pluronic F-127 gel. Approximately 200 freshly hatched second-stage juveniles (J2s) were gently mixed into the gel to ensure uniform distribution. The prepared dispenser was placed at the center of the dish and incubated at 28 °C for 20 h. Following incubation, 800 μL of distilled water was added to the dish, which was then maintained at 4 °C for 10 min to facilitate gel liquefaction. The dispenser was rinsed thoroughly, and nematodes were collected for enumeration. The same procedure was carried out for RNAi-treated nematodes. The numbers of nematodes located inside and outside the dispenser were recorded, and the attraction rate was calculated using the following formula: Attraction rate (%) = [Number of nematodes inside dispenser/(Number inside + Number outside)] × 100%.

### 2.4. Tomato Pot Experiment with Meloidogyne incognita

Tomato seedlings at the four-true-leaf stage from seedling trays were transplanted into plastic pots (height: 19 cm, diameter: 9 cm) and pre-cultured in a greenhouse at 25 °C for 11 days before nematode inoculation. Collected second-stage juveniles (J2s) were treated for 24 h in a soaking solution containing 500 μg/mL of target gene dsRNA. Nematodes treated in dsRNA-free soaking solution (Buffer) and non-targeting dsRNA (NC-dsRNA) served as controls. After treatment, the nematodes were thoroughly washed three times with sterile water to remove residual dsRNA. Approximately 200 J2s processed as described above were then manually inoculated around the roots of each tomato seedling at the four-leaf stage. After inoculation, the plants were maintained in the greenhouse for 60 days (25 °C, 50% relative humidity, 16/8 h light/dark cycle), after which the numbers of root galls and egg masses per plant root were counted for the different treatment groups.

### 2.5. Egg Hatching Assay of Meloidogyne incognita

The egg mass was placed into a 24-well plate. The plates were incubated at 28 °C, and the number of hatched second-stage juveniles (J2s) from each egg mass was recorded under a microscope(Olympus Corporation Shinjuku, Tokyo, Japan) at 12 h intervals.

### 2.6. Data Analysis

All experimental data were processed using GraphPad Prism 8.0 software. Statistical significance was analyzed by two-way ANOVA followed by Tukey’s multiple comparisons test. Each treatment had three parallel samples. All data were derived from three independent replicates, and a *p*-value < 0.05 was considered statistically significant.

## 3. Results and Discussion

### 3.1. RNAi of MAPK Genes in Meloidogyne incognita

Compared to the Buffer control and NC control groups, the expression levels of *Mi-mpk-1*, *Mi-mek-2*, *Mi-lin-45*, and *Mi-let-60* were significantly downregulated following RNA interference (Figure 1).

Under *Mi-mpk-1* silencing conditions, the expression levels of *Mi-mek-2* and *Mi-lin-45* were downregulated, with the latter showing significant reduction, while *Mi-let-60* expression was significantly upregulated compared to the Buffer control (Figure 2A). Following RNAi targeting *Mi-mek-2*, no significant downregulation was observed in *Mi-mpk-1* expression; however, *Mi-lin-45* and *Mi-let-60* exhibited marked upregulation (Figure 2B). When *Mi-lin-45* was silenced, its expression decreased as expected. Concurrently, *Mi-mpk-1* expression increased, whereas *Mi-mek-2* was significantly downregulated. *Mi-let-60* expression was also significantly elevated under these conditions (Figure 2C). In nematodes with silenced *Mi-let-60*, expression of the target gene was downregulated. Additionally, both *Mi-mpk-1* and *Mi-mek-2* showed significant downregulation, and *Mi-lin-45* displayed the most pronounced reduction in expression among all tested genes (Figure 2D).

The classical MAPK pathway consists of a three-tier kinase cascade (MAPKKK→MAPKK→MAPK), whose function relies on sequential activation and feedback regulation among its components. In *M. incognita*, Mi-LIN-45 (MAPKKK), Mi-MEK-2 (MAPKK), Mi-MPK-1 (MAPK), and the upstream signaling molecule *Mi-LET-60* (Ras homolog) constitute the RTK/Ras/MAPK signaling pathway. Experimental results demonstrate that the regulatory interactions among these genes are highly dynamic, with *Mi-let-60* (Ras) playing a central role. Silencing *Mi-let-60* led to significant downregulation of downstream *Mi-mpk-1* and *Mi-mek-2* (Figure 2D), indicating that Ras signaling drives the cascade by activating MAPKKK (*Mi-lin-45*). This observation aligns with the regulatory pattern of *let-60* in the MAPK pathway of *C. elegans* [7] However, a distinct finding in this study is that silencing *Mi-lin-45* resulted in the upregulation of *Mi-let-60*, suggesting that parasitic nematodes may maintain pathway stability through negative feedback regulation [26]

*Mi-mpk-1* (MAPK) exhibits closed-loop regulatory capability. Following *Mi-mpk-1* interference, the expression levels of both *Mi-lin-45* and *Mi-mek-2* were concurrently reduced (Figure 2A), implying that MAPK may exert retrograde regulation on its upstream components through phosphorylation of transcription factors or non-coding RNAs, thereby forming a self-regulatory network [27] Such a feedback mechanism has been rarely documented in model organisms and may represent an adaptive strategy employed by *Meloidogyne* spp. to maintain signaling homeostasis in complex host environments. These findings provide evidence of the cascade nature and complexity of the MAPK pathway in *Meloidogyne* spp., offering new insights into the signal transduction networks in parasitic nematodes.

### 3.2. Chemotactic Activity of Meloidogyne incognita After RNAi

Compared to the Buffer control and NC control groups, the attraction rate of 0.1 M citric acid to *M. incognita* J2s was reduced to varying degrees following RNA interference of the *Mi-mpk-1*, *Mi-mek-2*, *Mi-lin-45*, and *Mi-let-60* genes. Notably, silencing of *Mi-mpk-1* and *Mi-let-60* resulted in a 40–50% decrease in the chemotactic response of *Meloidogyne* spp. to citric acid (Figure 3).

Compared to the Buffer control and NC control groups, tomato plants inoculated with *Meloidogyne incognita* subjected to RNAi targeting *Mi-mpk-1*, *Mi-mek-2*, or *Mi-let-60* showed a 70.01–87.67% reduction in root gall and a 31.60–71.86% decrease in egg mass. In contrast, silencing of the *Mi-lin-45* gene did not result in a statistically significant decrease in either gall number or egg mass count compared to the NC control group.

With regard to the regulation of chemotaxis, previous studies have predominantly focused on the perception of volatile compounds. For example, Hirotsu [7] demonstrated that mutations in *let-60* in *C. elegans* impair the perception of volatile attractants such as isoamyl alcohol. The present study extends these findings by showing that RNA interference of *Mi-let-60* or *Mi-mpk-1* in *M. incognita* significantly reduced the nematode’s chemotactic response to the water-soluble compound citric acid by 40–50% (Figure 3A). This indicates that the MAPK signaling pathway is involved in integrating diverse chemical 0ues, including non-volatile, water-soluble signals. These results provide novel insights into the molecular mechanisms underlying interactions between root exudates and plant-parasitic nematodes. Furthermore, in comparison to the 32% reduction in root gall formation reported by Dong [19] following single-gene silencing of *mpk-1*, our approach involving multi-gene interference achieved a greater than 70% reduction in galling (Figure 4), suggesting that simultaneous targeting of multiple components within the MAPK pathway can markedly enhance the efficacy of nematode control.

### 3.3. Effect of MAPK Genes on Egg Hatching Ability in Meloidogyne incognita

Egg masses of *M. incognita* were subjected to RNAi interference targeting *Mi-mpk-1*, *Mi-mek-2*, *Mi-lin-45*, and *Mi-let-60*, respectively. Using buffer and NC as control groups, the expression levels of these MAPK pathway genes were quantified at 24, 48, and 72 h post-treatment (Figure 5). The results show that *Mi-mpk-1*, *Mi-mek-2*, and *Mi-lin-45* were effectively silenced in the egg masses. Although the expression level of *Mi-let-60* exhibited a declining trend across the three time points following RNAi, its overall expression remained significantly higher than in the NC and buffer control groups. Notably, the transcript levels of *Mi-mek-2* and *Mi-lin-45* showed partial recovery at 72 h after RNAi treatment.

Egg masses of *M. incognita* were individually subjected to RNA interference targeting *Mi-mpk-1*, *Mi-mek-2*, *Mi-lin-45*, and *Mi-let-60*. The number of hatched J2s in each treatment group was evaluated at 24, 48, and 72 h post-treatment, with buffer and NC groups serving as controls. Silencing of *Mi-lin-45* and *Mi-let-60* resulted in a reduction in J2 hatch numbers compared to the buffer control. Notably, egg masses treated with *Mi-let-60* RNAi exhibited a significant decline in hatching capacity by 72 h. In contrast, interference with Mi-mpk-1 led to increased J2 counts at 24 and 48 h relative to the control, but a sharp decrease was observed at 72 h, resulting in final hatch counts below those of the control group. For *Mi-mek-2*-silenced egg masses, hatching rates remained comparable to controls at 24 h but dropped below control levels after 48 h. Although both *Mi-mek-2* (MAPKK) and *Mi-lin-45* (MAPKKK) are components of the same MAPK signaling cascade, their roles in regulating egg hatching appear distinct. While knockdown of *Mi-lin-45* only mildly suppressed hatching, silencing of *Mi-mek-2* caused delayed hatching, suggesting that MAPKK may directly regulate embryonic cell division through modulation of cell cycle-associated proteins such as CDK1 [6].

With regard to egg hatching [18], previously reported that RNA interference of *Mi-mpk-1* increased hatching rates at early stages, although the long-term consequences were not fully elucidated. Our study not only confirms this observation but also reveals a significant decline in hatching rates after 72 h (Figure 6). We propose that the MAPK pathway may regulate the balance between short-term survival and long-term development through a “stress-compensation” mechanism. For example, transient gene silencing might activate compensatory signaling pathways such as p38 or JNK, thereby promoting embryonic escape under stress conditions, whereas sustained disruption could lead to a deficiency in essential developmental signals [28]. Moreover, silencing of *Mi-let-60* markedly suppressed egg hatching, reducing the hatching rate by approximately 50% at 72 h. Although the role of *Mi-let-60* in the embryonic development of *Meloidogyne* spp. has not been previously documented, our findings indicate that Ras signaling may serve as a novel regulatory target for controlling egg hatching in *Meloidogyne* spp.

## 4. Conclusions

This study systematically analyzed the functions of four key genes (*Mi-mpk-1*, *Mi-mek-2*, *Mi-lin-45*, and *Mi-let-60*) in the MAPK pathway of *Meloidogyne incognita* in chemotaxis and egg hatching using RNAi technology. The most significant findings include the revelation of a unique feedback regulation mechanism within the MAPK pathway and the functional divergence and synergy among these genes. Specifically, silencing *Mi-lin-45* (MAPKKK) led to a significant upregulation of the upstream gene *Mi-let-60* (Ras), suggesting the presence of a potential negative feedback mechanism in root-knot nematodes to maintain signaling homeostasis, which may represent an adaptive strategy for survival in fluctuating host environments. Concurrently, this study revealed clear functional divergence among these genes: *Mi-let-60* and *Mi-mpk-1* were identified as critical nodes regulating host location in chemotaxis, whereas *Mi-mek-2* and *Mi-mpk-1* played more direct roles than *Mi-lin-45* in egg hatching. Crucially, regarding pathogenicity, the simultaneous interference of *Mi-let-60* and *Mi-mpk-1* produced a synergistic effect, resulting in a drastic reduction (over 70%) in root galls and egg masses, which was far superior to single-gene interference, highlighting the great potential of multi-target synergistic intervention for disease control. These results not only confirm the conservation of the MAPK pathway in *M. incognita* but also uncover its unique internal feedback regulation and functional divergence patterns, which collectively optimize the parasitic adaptability of the nematode. This study provides a solid theoretical foundation for developing green control strategies targeting multiple nodes of the MAPK pathway.

## Figures and Tables

**Figure 1 pathogens-14-01290-f001:**
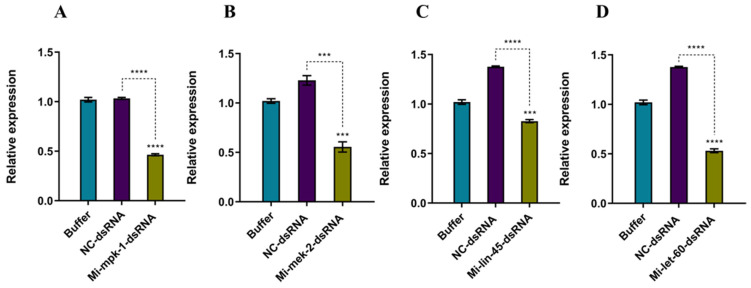
**Relative expression levels of MAPK genes in *M. incognita* J2s after RNAi.** Buffer: soaking solution without dsRNA; NC-dsRNA: negative control with non-target dsRNA fragment. (**A**) Relative expression level after RNAi of *mpk-1*; (**B**) Relative expression level after RNAi of *mek-2*; (**C**) Relative expression level after RNAi of *lin-45*; (**D**) Relative expression level after RNAi of *let-60*; Bars: SEM. (two–way ANOVA, Tukey’s multiple comparisons, *** *p* < 0.001, **** *p* < 0.0001).

**Figure 2 pathogens-14-01290-f002:**
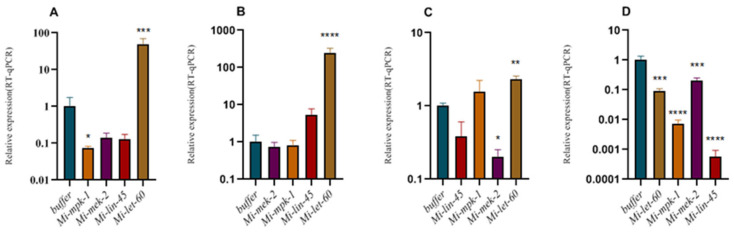
**Relative expression levels of non-target genes after RNAi of individual target genes.** Buffer: soaking solution without dsRNA. (**A**) Relative expression levels of non-target genes after RNAi of *mpk-1*; (**B**) Relative expression levels of non-target genes after RNAi of *mek-2*; (**C**) Relative expression levels of non-target genes after RNAi of *lin-45*; (**D**) Relative expression levels of non-target genes after RNAi of *let-60*; Bars: SEM. (two–way ANOVA, Tukey’s multiple comparisons, * *p* < 0.05, ** *p* < 0.01, *** *p* < 0.001, **** *p* < 0.0001).

**Figure 3 pathogens-14-01290-f003:**
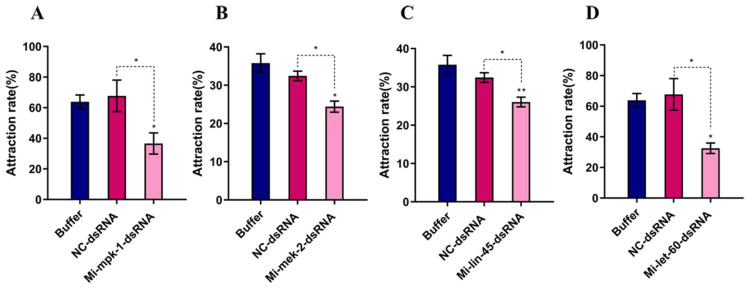
**Chemotactic response of *M. incognita* J2s to citric acid after RNAi.** Buffer: soaking solution without dsRNA; NC-dsRNA: negative control with non-target dsRNA fragment. (**A**) Chemotactic activity of *Meloidogyne incognita* J2s toward citric acid after RNAi of *mpk-1*; (**B**) Chemotactic activity of *Meloidogyne incognita* J2s toward citric acid after RNAi of *mek-2*; (**C**) Chemotactic activity of *Meloidogyne incognita* J2s toward citric acid after RNAi of *lin-45*; (**D**) Chemotactic activity of *Meloidogyne incognita* J2s toward citric acid after RNAi of *let-60*; Bars: SEM. (two–way ANOVA, Tukey’s multiple comparisons, * *p* < 0.05, ** *p* < 0.01).

**Figure 4 pathogens-14-01290-f004:**
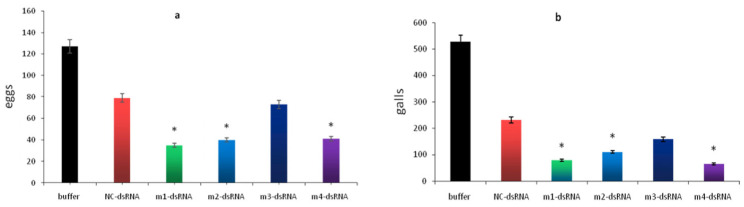
**Infection assay of tomato by nematodes after RNAi treatment.** (**a**):Number of egg masses formed by nematodes after RNAi treatment of *mpk-1*, *mek-2*, *lin-45*, and *let-60* genes; (**b**) Number of galls formed by nematodes after RNAi treatment of *mpk-1*, *mek-2*, *lin-45*, and *let-60* genes. Bars: SEM. (two–way ANOVA, Tukey’s multiple comparisons, * *p* < 0.05).

**Figure 5 pathogens-14-01290-f005:**
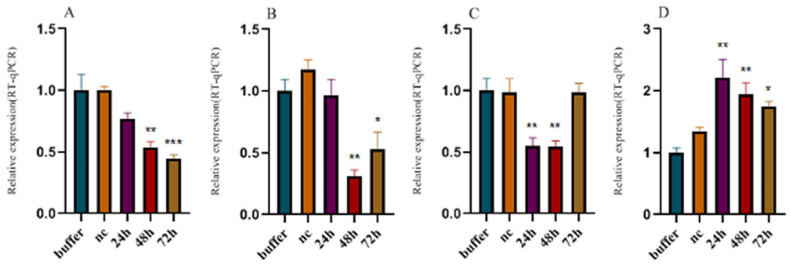
**Gene expression levels of *M. incognita* egg masses after RNAi treatment.** Buffer: soaking solution without dsRNA; NC-dsRNA: negative control with non-target dsRNA fragment. (**A**) Expression levels of egg-mass-related genes in *Meloidogyne incognita* at 24, 48, and 72 h after RNAi of *mpk-1*; (**B**) Expression levels of egg-mass-related genes in *Meloidogyne incognita* at 24, 48, and 72 h after RNAi of *mek-2*; (**C**) Expression levels of egg-mass-related genes in *Meloidogyne incognita* at 24, 48, and 72 h after RNAi of *lin-45*; (**D**) Expression levels of egg-mass-related genes in *Meloidogyne incognita* at 24, 48, and 72 h after RNAi of *let-60*. Bars: SEM. (two–way ANOVA, Tukey’s multiple comparisons, * *p* < 0.05, ** *p* < 0.01, *** *p* < 0.001).

**Figure 6 pathogens-14-01290-f006:**
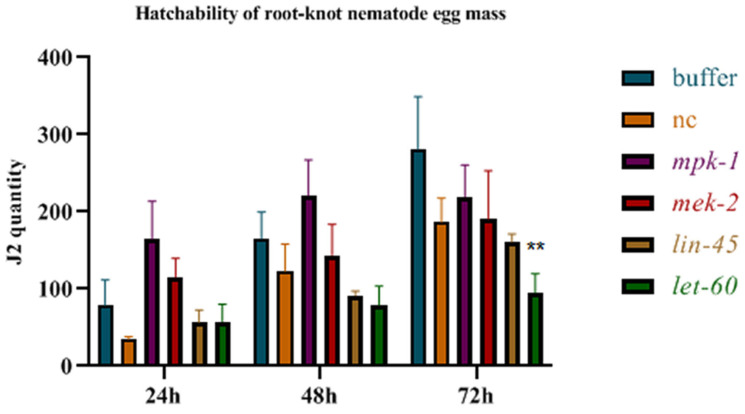
**Alterations in egg hatching capacity of *M. incognita* after RNAi treatment.** Buffer: soaking solution without dsRNA; NC-dsRNA: negative control with non-target dsRNA fragment. Bars: SEM. (two–way ANOVA, Tukey’s multiple comparisons, ** *p* < 0.01).

## Data Availability

The original contributions presented in this study are included in the article/Appendix A. Further inquiries can be directed to the corresponding author.

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
