# Peer review of "Effects of MAPK Homologous Genes on Chemotaxis and Egg Hatching in Meloidogyne incognita"

_pathogens, 2025, doi:10.3390/pathogens14121290_

Round 1

Reviewer 1 Report

Comments and Suggestions for Authors

An interesting study was provided to reveal a functional characterization of MAPK in M. incognita. This manuscript provides a valuable insight yet requires some points to be trimmed as following.

  1. line 99, chemotaxis usually has a target molecule. Could you briefly introduce which are the representative target molecules?
  2. Some of the species names (such as in line 110) are not italic.
  3. Line 184~186. The description is too brief. Please elaborate which approach was taken for which question, and what was the experimental condition at least in minimal level for the readers (I know it is in M&M but most of the readers need this in the result section.)
  4. Please rewrite the figure legends to be more kind. I was not able to understand. e.g. In the figure 1, A: mpk-1 indicates a target gene? Then, what is annotated in X-axis?
  5. Throughout the study, you are using RNAi treated M. incognita for the physiological experiments. Are the physiological or molecular experiments and RNAi treatment done in the same day or different day / trial? If it is the later, the individual RNAi treatment should be confirmed.
  6. The detail for infection assay is required. With the line 241-245, I was not able to recognize the RNAi was treated after inoculation or before inoculation.
  7. As this is an RNAi treatment analysis, it seems the average half-life of the RNAi molecules are important to assessing Figure 5. Did authors assessed this point?
  8. Throughout the paper, the replicate number and characteristic is missing. Was it triplicate? Were they performed in biological or technical manner?
  9. Was egg hatching assay performed in vitro or performed with plant roots? By the detailed condition, I think the biological meaning will be more expressed and could be emphasized.

Reviewer 2 Report

Comments and Suggestions for Authors

The manuscript "Effects of MAPK Homologous Genes on Chemotaxis and Egg Hatching in Meloidogyne incognita" addresses the actual problem of identification of MAPK signaling role in the development and responses of root-knot nematodes. The work was carried out using modern research methods, manuscript is well-structured. However, I would like to make several recommendations to the authors for improving the presentation of the material.

Key comments:

  1. Abstract. The authors should make this section more specific by including the most important results obtained in the work. The abstract should be readable independently of the main text of the manuscript.
  2. Materials and Methods section, subsection "Culture and Collection of Meloidogyne incognita".

The subsection title should be corrected, as it describes the growing and treatment conditions of the plants.

  1. The same subsection, line 115. In addition to the greenhouse temperature, the watering regime and, preferably, the relative humidity in the greenhouse should be indicated.
  2. Authors should indicate the source of the M. incognita J2s used in the work. It should also specify how the species was identified – by morphological or molecular methods.
  3. The subsection "Tomato Pot Experiment with Meloidogyne incognita" repeats lines 112-118 and lines 130-132.
  4. In the "Data Analysis" subsection, please, indicate the number of independent experiments (n).
  5. Conclusion. Authors should specify which "distinct gene interaction patterns and their novel contributions to parasitic adaptation" (lines 319-320) were identified in the study. It is crucial that the reader be able to obtain concise, specific information about the most important results of the study from the Conclusion.
  6. Authors should correct the formatting of the reference list so that it complies with the journal's requirements.

Minor comments.

Authors should carefully check the text for the correct using of Latin names, abbreviations, and the presence of spaces between words. Abbreviations of terms should be given the first time the term is mentioned in the text, and then only the abbreviated version should be used. When using Latin names, the full name should be given the first time it is mentioned in the text, followed by the abbreviated version. Latin names should be written in italics.

For example:

  1. Line 36 – please, add a space after "survival";
  2. Line 38 – please, add a space after "C. elegans";
  3. Line 40 – please, add a space after "(Enk)";
  4. Line 57 – please, add a space after "odorants" and before "Genetic";
  5. Line 75 – the full name of "MAPK" was given earlier, on line 33. Do not repeat it, please.
  6. Line 82 – please, add a space before "cloned";
  7. Lines 83, 87, 91 – in vitro should be written in italics;
  8. Line 96 – M. incognita should be written in italics;
  9. Line 100 – the full name of "J2s" was given earlier, on line 86; analogical comment for lines 123 and 285-286;
  10. Line 110 – Solanum lycopersicum – should be written in italics;
  11. Line 212 – Meloidogyne incognita should be written in italics and in abbreviated form (M. incognita); similarly – line 284;
  12. Line 251 – yellow highlighting should be removed;
  13. Line 266 – the explanation that NC stands for non-targeting dsRNA was given earlier in the text and should not be repeated; similarly – line 287;
  14. Line 300 – a reference to the source (Chen et al., 2008) should be added.

The above comments do not reduce the practical and fundamental value of the work and are aimed only at improving the quality of the manuscript presentation and its perception by readers.

Reviewer 3 Report

Comments and Suggestions for Authors

Dear authors, 

I would like to congratulate you on this highly relevant research. I only recommend minor changes aimed to improving the understanding of the work. 

The suggestions are described below:

Please put the scientific names in italics (Lines 24, 110 e 284).

Material and methods: 

Describe how the dsRNAs were prepared and which genes were targeted (item line 121).

How was the RT-PCR analysis performed?

Reference number 20 does not cover a specific protocol. Please include specific references for the methods used.

Round 2

Reviewer 1 Report

Comments and Suggestions for Authors

I think authors have sincerely provided a revised manuscript fully reflecting the reviewer's comment. Yet, when you publish this article, please upgrade the figure dpi.

Reviewer 2 Report

Comments and Suggestions for Authors

The authors took all my suggestions into account and significantly revised the manuscript text. However, several points remain that require correction:
1. The reference list still needs to be corrected. Follow the the Instructions for Authors: Abbreviated Journal Name, Year, Volume. Journal and volume number in italics, year in bold.

Author Response

1.The reference list still requires correction. Please follow the Instructions for Authors: Journal name in abbreviated form and italics, year in bold, and volume number in italics.

reponse:The reference list has been corrected according to the specified format. All entries now adhere to the Instructions for Authors: abbreviated journal names and volume numbers are in italics, and publication years are in bold.